# The Effects of *L*-citrulline Supplementation on the Athletic Performance, Physiological and Biochemical Parameters, Antioxidant Capacity, and Blood Amino Acid and Polyamine Levels in Speed-Racing *Yili* Horses

**DOI:** 10.3390/ani14162438

**Published:** 2024-08-22

**Authors:** Peiyao Li, Shuo Sun, Wenjie Zhang, Wen Ouyang, Xiaobin Li, Kailun Yang

**Affiliations:** 1Xinjiang Key Laboratory of Herbivore Nutrition for Meat & Milk Production, College of Animal Science, Xinjiang Agricultural University, Urumqi 830052, China; 16699218860@163.com (P.L.); s1161931970@163.com (S.S.); xjauzwj@126.com (W.Z.); 2Yili Kazak Autonomous Prefecture of Zhaosu Racecourse, Yining 835000, China; xinxin961029@163.com

**Keywords:** *L*-citrulline, *Yili* horse, free amino acid, NO, polyamines, athletic performance

## Abstract

**Simple Summary:**

*L*-citrulline, as a non-essential amino acid, is recognized for its effectiveness in enhancing arginine bioavailability and nitric oxide (NO) synthesis, drawing significant attention from the scientific community for its potential to improve athletic performance. This study investigated the effects of *L*-citrulline supplementation on the racing performance of *Yili* horses as well as on the blood acid–base balance, physiological and biochemical blood indices, and antioxidant markers before and after an exercise. Additionally, changes in the plasma amino acid metabolism and polyamine levels were examined. The results indicate that the supplementation of 50 g of *L*-citrulline to the horses’ diet significantly increased the plasma concentrations of citrulline and arginine and provided a certain improvement in the athletic performance of the *Yili* horses. Additionally, compared to a control group, the total protein and lactate dehydrogenase levels were significantly elevated 2 h before and 2 h after the race, while the lactate concentration immediately post-race was significantly reduced.

**Abstract:**

The objective of this study was to evaluate the effects of pre-exercise *L*-citrulline supplementation on the athletic performance of *Yili* speed-racing horses during a high-intensity exercise. On the 20th day of the experiment, blood samples were collected at 3 h and 6 h post-supplementation to measure the amino acid and polyamine concentrations. On the 38th day of the experiment, the horses participated in a 2000 m speed race, and three distinct blood samples were gathered for assessing blood gases, hematological parameters, the plasma biochemistry, antioxidant parameters, and NO concentrations. The results indicate that the *L*-citrulline group showed a significant increase in the plasma citrulline and arginine concentrations. Conversely, the concentrations of alanine, serine, and threonine were significantly decreased. The glycine concentration decreased significantly, while there was a trend towards an increase in the glutamine concentration. Additionally, the levels of putrescine and spermidine in the plasma of the *L*-citrulline group were significantly increased. In terms of exercise performance, *L*-citrulline can improve the exercise performance of sport horses, significantly reduce the immediate post-race lactate levels in *Yili* horses, and accelerate the recovery of blood gas levels after an exercise. Furthermore, in the *L*-citrulline group of *Yili* horses, The levels of the total protein of plasma, superoxide dismutase, catalase, and lactate dehydrogenase were significantly increased both 2 h before and 2 h after the race. The total antioxidant capacity showed a highly significant increase, while the malondialdehyde content significantly decreased. In the immediate post-race period, the creatinine content in the *L*-citrulline group significantly increased. In conclusion, this study demonstrates that *L*-citrulline supplementation can influence the circulating concentrations of *L*-citrulline and arginine in *Yili* horses, enhance the antioxidant capacity, reduce lactate levels, and improve physiological and biochemical blood parameters, thereby having a beneficial effect on the exercise performance of athletic horses.

## 1. Introduction

During high-intensity exercises, the contraction of muscle cells in horses leads to energy consumption and an increased oxygen demand, resulting in significant changes in metabolic and physiological states, which can affect the health of horses. Therefore, the development of new nutritional supplements is of great importance for replenishing energy in skeletal muscles and enhancing the oxygen-uptake capacity of performance horses. *L*-citrulline is a non-essential, non-coding amino acid and an effective precursor for the synthesis of arginine [1]. Studies have shown that *L*-citrulline supplementation is more effective at increasing plasma arginine levels compared to the direct oral intake of arginine [2]. Therefore, the inclusion of *L*-citrulline in feed can circumvent the limitations on arginine availability during intense exercises. Arginine has been shown to significantly improve physiological functions in various species and serves as a precursor for numerous substances in animal metabolism, including NO, polyamines, and creatine [3]. However, the direct intake of arginine results in substantial absorption by the liver from portal blood, reducing its availability. In contrast, *L*-citrulline bypasses the liver and is metabolized in the kidneys [4]. Additionally, there is limited research on the effects of increased *L*-citrulline intake on exercise metabolism in exercising horses or on the impact of combining exercise training with *L*-citrulline supplementation on *L*-citrulline absorption and changes in free amino acids. Studies suggest that the exogenous supplementation of *L*-citrulline may increase NO concentrations [5]. In exercise physiology, NO can enhance muscle contraction efficiency, improve exercise endurance, regulate oxygen consumption [6], and aid in post-exercise recovery [7], thereby altering skeletal muscle metabolism during exercises. This is particularly significant for athletes, as they often have a limited recovery time due to the high frequency of training and competition. Some studies have indicated that citrulline can increase blood flow to the muscles, leading to the enhanced delivery of oxygen and nutrients, which in turn boosts the muscle energy supply and improves exercise performance [8]. Currently, *L*-citrulline has been studied as a sports nutrition supplement in both athletes and mice. Research has found that the long-term supplementation of *L*-citrulline in healthy male athletes positively affects aerobic exercise performance and training-induced skeletal muscle adaptations [9]. In mice, supplementation with *L*-citrulline prior to exhaustive swimming significantly extends the swimming time and enhances the exercise-induced oxidative stress capacity [10]. Data indicate that higher doses of *L-*citrulline may have a more pronounced effect on improving exercise performance [11]. Therefore, the aim of this study is to investigate the effects of *L-*citrulline supplementation (50 g/d) on the exercise performance and metabolism of *Yili* horses during a 2000 m speed race. By analyzing changes in the athletic performance, plasma amino acid concentrations, physiological and biochemical parameters, antioxidant capacity, and blood gas indices before and after the race, this study seeks to determine whether *L-*citrulline supplementation can increase the plasma concentrations of *L-*citrulline and arginine, enhance the horses’ 2000 m race performance, and improve their resistance to exercise-induced stress.

## 2. Materials and Methods

### 2.1. Trial Date and Location

The present study was conducted in Zhaosu *Yili* Kazak Autonomous Prefecture over a 39-day period between September and October 2022. All animal care and handing procedures in this study were conducted under the Guidance of the Care and Use of Laboratory Animals in China and were approved by the Animal Care Committee of the Xinjiang Agricultural University of China (protocol permit number: 2020024).

### 2.2. Animal and Experimental Design

In this experiment, twelve *Yili* horses aged 2 years, with an average body weight of (407.12 ± 36.91) kg and similar 2000 m race times, were selected as the study subjects. They were randomly divided into two groups, with six horses in each group (5 males and 1 female), designated as the control group and *L*-citrulline group. Throughout the entire experiment, all experimental horses were provided with a consistent diet composition and nutritional levels. Each horse was fed 8 kg of dried forage and 6 kg of a concentrate supplement daily, with free access to water. Based on Daniel’s study [12], the *L*-citrulline group received an additional 50 g of *L*-citrulline mixed thoroughly into the concentrate supplement for each horse daily and was supplemented once in the morning and once in the evening with 25 g each time, and the feeding time was at 8:30 and 20:30 every day, respectively. The experimental horses underwent daily training of equal intensity and participated in a 38-day supplementation training trial. On the 38th day, a 2000 m speed race was conducted. All horses were housed individually in separate stalls, which were cleaned daily to remove manure and bedding, and fresh dry straw was provided. The dry forage was fed four times daily at 8:00, 13:00, 19:00, and 23:00. Each time, a moderate amount of dry forage (approximately 1.0 kg) was offered first. After the dry forage was consumed, the concentrate supplement was provided. Once the concentrate supplement was eaten, the remaining dry forage was made available for free access. The composition and nutritional content of the concentrate supplement are shown in Table 1, while the nutritional content of the dried forage is presented in Table 2.

After consuming the forage, the horses were allowed to exercise freely in the outdoor activity area. All test horses underwent speed training once daily, with the training sessions being scheduled from 11:00 to 13:00 or 16:00 to 18:00. The training routine began with the horses being saddled and led to the training ground. The training started with a 10 min warm-up at a slow trot on the grass, followed by a gradual increase in speed to a brisk trot on the sand for another 10 min. The training concluded with 15 min of running, followed by 5–10 min of leading the horses until their breathing stabilized.

### 2.3. Sample Collection and Analysis

On the 38th day of the trial, a 2000 m speed race was conducted at the Zhaosu County Stallion Farm Test and Conditioning Centre. The race was overseen by two judges, three timekeepers, and one data recorder, with 12 professional jockeys selected, all with similar average body weights (65 ± 3.5 kg). A stopwatch is used by the timekeeper to accurately determine the test horse’s race time during the race. Upon verification by the judges, the average value of the valid data was taken as the performance result of the racehorses. Respiratory rates were measured using a stethoscope in the horses’ chest at the same intervals: 2 h before the race, immediately after the race, and 2 h post-race. Additionally, Heart rates of all horses were measured using Polar heart rate monitors (Yilian Scientific and Educational Equipment Co., Ltd., Shanghai China) 2 h before the race, immediately after the race, and 2 h post-race.

For this experiment, blood samples were collected from the jugular vein of all horses.

On the 20th day of the experiment, 5 mL of blood was collected 3 h post-supplementation to determine the plasma amino acid levels. Additionally, 2 mL of blood was collected 6 h post-supplementation and immediately mixed with 2 mL of 15% perchloric acid using a pipette to measure the plasma polyamine concentrations.

On the 38th day of the experiment, a 2000 m speed race was conducted. Blood samples were collected 2 h before the race, immediately after the race, and 2 h post-race to measure the blood gas parameters, physiological and biochemical indices, antioxidant markers, and nitric oxide (NO) levels. Blood samples of 2 mL were collected using standard blood-collection tubes and EDTA-K anticoagulant tubes, and the blood gas and routine blood parameters were measured immediately. Additionally, 5 mL of blood was collected using sodium heparin anticoagulant tubes.

These samples were immediately centrifuged at 3500 r/min for 10 min to separate the plasma, which was then labeled and stored at −80 °C for future analysis.

### 2.4. Sample Determination

#### 2.4.1. Blood Gas Parameter

A real-time measurement of freshly collected blood samples was performed immediately using the i-SATA portable blood gas analyzer (with Ca18 test strips) to determine the blood gas parameters. The measured indices include the blood pH; partial pressure of oxygen (PO_2_); partial pressure of carbon dioxide (PCO_2_); and electrolyte concentrations such as Na^+^, K^+^, and ionized calcium (iCa^2+^). The biochemical indices include blood glucose (Glu), lactate (Lac), and hematocrit (Hct). The pH and PCO_2_ levels were measured using the voltage method; PCO_2_, Glu, and Lac were measured using the current method; Na^+^ and K^+^ were measured using the electrochemical method; iCa^2+^ was measured using the selective electrode voltage method; and Hct was measured using the electrical conductivity method. For the determination of blood gas indicators, refer to the study by Li et al. [13].

#### 2.4.2. Hematologic Indices

The hematologic indices were determined by the BC-5300 Vet Mindray automatic five-taxonomic animal blood cell analyzer (Tianjin Mnc Technologies Co., Ltd., Tianjin China). The measures included white blood cells (WBCs), lymphocytes (Lyms), neutrophils (Neus), monocytes (Mons), eosinophils (Eos), basophils (Bas), red blood cells (RBCs), and platelets (PLTs). The determination of hematologic indices was performed with reference to Li [14] and Yang et al. [15].

#### 2.4.3. Plasma Biochemical Indices

The total protein (TP), albumin (Alb), globulin (Glb), creatine kinase (CK), lactate dehydrogenase (LDH) activity, creatinine (Cre), and urea (Urea) in the plasma were evaluated using commercial assay kits (Beijing Huaying Biotechnology Research Institute, Beijing China). In accordance with the manufacturers’ guidelines, all protocols were strictly carried out, ensuring the highest level of safety and accuracy. The concentrations of TP, Alb, Glb, CK, LDH, Cre, and Urea in the plasma were measured using a 7230 G spectrophotometer by the colorimetric method.

#### 2.4.4. Plasma Antioxidant Capacity and Determination of NO Concentrations

The concentrations of the plasma nitric oxide (NO, HY-60018); total antioxidant capacity (T-AOC, HY-60021); and malondialdehyde (MDA, HY-M0003) and the activities of superoxide dismutase (SOD, HY-M0001); catalase (CAT, HY-M0018); and glutathione peroxidase (GSH-Px, HY-M0004) were measured using the colorimetric method. All samples were sent to the Beijing Huaying Biotechnology Research Institute for analysis [16].

#### 2.4.5. Plasma Amino Acid and Polyamine Content

The plasma concentrations of the amino acids and polyamines were analyzed and detected by Purui Huasheng Medical Laboratory Co., Ltd. (Tianjin, China) using high-performance liquid chromatography–tandem mass spectrometry (HPLC-MS/MS). The instrument model used was the Shimadzu LC20AD—API 3200MD TRAP (Shimadzu Enterprise Management Co., Ltd., Shanghai, China).

Amino acid derivatization procedure (Figure 1): A 50 µL mixture of a standard mix and the sample to be tested was added to 50 µL of a protein precipitant (10% sulfosalicylic acid containing NVL), and the mixture was thoroughly mixed and centrifuged at 13,200 rpm for 4 min at a low temperature. An 8 µL aliquot of the supernatant was mixed with 42 µL of a labeling buffer (borate buffer, pH = 8.5) and briefly centrifuged. Subsequently, 20 µL of the derivatization reagent AQC was added, and the mixture was thoroughly mixed, briefly centrifuged, and then incubated at 55 °C for 15 min for derivatization. The derivatized samples were then cooled in a refrigerator and briefly centrifuged, and 50 µL of the sample was taken for analysis using an instrument.

MSLAB-45 + AA (Purui Huasheng Medicine Laboratory Co., Ltd. (Tianjin, China). Methanol and acetonitrile were purchased from the American Sigma Corporation (St. Louis, MO, USA). The chromatographic and mass spectrometry conditions of amino acids are shown in Table 3. The standard curves and peak times of amino acids are shown in Table 4.

Polyamine derivatization treatment: A liquid sample of 100 µL was taken and added to 100 µL of a buffer solution consisting of sodium carbonate (5 mol/L) and sodium bicarbonate (0.5 mol/L), with a pH of 11.5. Then, 100 µL of dansyl chloride (10 mg/mL) was added and vortexed for 1 min. After brief centrifugation, the mixture was incubated at 60 °C for 15 min to allow for derivatization, then cooled to room temperature. Subsequently, 800 µL of an extraction solution consisting of n-hexane and ethyl acetate (8:2) was added, vortexed for 5 min, and centrifuged at 13,200 rpm for 6 min at a low temperature. Finally, a 600 µL aliquot of the supernatant was taken and dried under nitrogen. The residue was reconstituted with 100 µL of acetonitrile, vortexed for 1 min, and centrifuged at 13,200 rpm for 6 min at a low temperature. An 80 µL aliquot of the sample was then collected for analysis.

Measurement principle: R1R2-NH2 + DNS-Cl → R1R2NH-DNS + HCl.

Putrescine, spermidine, and spermine were obtained from the Purui Huasheng Medicine Laboratory Co., Ltd. (Tianjin, China). Methanol and acetonitrile were purchased from the American Sigma Corporation. The chromatographic and mass spectrometry conditions of polyamines are shown in Table 5. The standard curves and peak times of the polyamines are shown in Table 6.

### 2.5. Statistical Analysis

First, all data were sorted in Microsoft Excel 2010 (Microsoft Corp., Redmond, WA, USA). The subsequent analysis was performed using the SAS 9.4 software (Cary, NC, USA), specifically an independent samples *t*-test was conducted for the plasma amino acid and polyamine indices. Finally, the statistical analysis and plotting were conducted using the GraphPad Prism 9.5.1 software (La Jolla, CA, USA). The obtained experimental data are all expressed as a mean ± standard deviation, with *p* < 0.05 indicating significant differences and 0.05 < *p* < 0.10 indicating a significant trend of differences.

## 3. Results

### 3.1. Effects of L-citrulline Supplementation on Race Time, Respiration, and Heart Rate of Yili Horses before and after 2000 m Speed Race

As shown in Figure 2A, compared to day 0 of the experiment, the *L*-citrulline group’s race time (217.2 vs. 210 s) on day 38 decreased by 7.2 s, while the control group’s race time (219.8 vs. 215.4 s) decreased by 4.4 s. On day 38 of the experiment, both the *L*-citrulline and the control groups showed performance improvements. The *L*-citrulline group had a 2.51% reduction in the race time (210 vs. 215.4 s) compared to the control group.

As shown in Figure 2B,C, 2 h before the race, the physiological conditions of the horses in the *L*-citrulline and control groups were stable, with no significant differences in the respiratory rate (25.5 vs. 26.67 bpm, *p* = 0.192) and heart rate (39.33 vs. 40.17 bpm, *p* = 0.309). Immediately after the race, both the respiratory rate (84.17 vs. 83 bpm, *p* = 0.246) and heart rate (90.67 vs. 91.67 bpm, *p* = 0.692) increased in the *L*-citrulline and control groups, but the differences were not statistically significant. At 2 h post-race, the respiratory rate (28.5 vs. 29 bpm, *p* = 0.646) and heart rate (40.17 vs. 42.5 bpm, *p* = 0.09) in both the *L*-citrulline and control groups returned to normal levels. Additionally, there was a trend towards a lower heart rate in the *L*-citrulline group compared to the control group.

### 3.2. Effects of L-citrulline Supplementation on Blood Gas Indices of Yili Horses before and after 2000 m Speed Race

As shown in Figure 3, immediately after the race, the Lac concentration in the *L*-citrulline group (15.29 vs. 16.3 mmol/L, *p* = 0.042) was significantly lower than that in the control group, by 6.20%, while the Hb concentration (21.75 vs. 19.87 g/dL, *p* = 0.021) was significantly higher by 9.46%. The blood glucose concentration (204.5 vs. 199.8 mg/dL, *p* = 0.174) was higher than the control group, by 2.35%. A total of 2 h before the race, the blood glucose concentration in the *L*-citrulline group (103.3 vs. 99.83 mg/dL, *p* = 0.158) was 3.48% higher than that in the control group, while the Lac concentration (0.69 vs. 0.72 mmol/L, *p* = 0.254) was 4.17% lower than that in the control group. A total of 2 h after the race, the blood glucose concentration in the *L*-citrulline group (104 vs. 96.83 mg/dL, *p* = 0.347) was 7.40% higher than that in the control group, while the Lac concentration (2.38 vs. 2.64 mmol/L, *p* = 0.132) was 9.85% lower than that in the control group.

The plasma concentrations of the pH (7.48 vs. 7.48), PCO_2_ (39.6 vs. 38.1 mmHg), TCO_2_ (30.73 vs. 30.02 mmHg), and Ca^2+^ (1.64 vs. 1.67 mmol/L) in the *L*-citrulline and control groups 2 h before the race; as well as the plasma concentrations of the pH (7.22 vs. 7.17), PCO_2_ (25.63 vs. 21.6 mmHg), TCO_2_ (11.65 vs. 11.23 mmHg), and Ca^2+^ (1.47 vs. 1.48 mmol/L) immediately after the race; and the concentrations of the pH (7.46 vs. 7.44), PCO_2_ (40.72 vs. 43.05 mmHg), TCO_2_ (29.85 vs. 30.06 mmHg), and Ca^2+^ (1.67 vs. 1.68 mmol/L) 2 h after the race all exhibited a trend of initially decreasing and then increasing throughout the entire period. The plasma concentrations of the PO_2_ (35.3 vs. 35.03 mmHg) and Hct (37.33 vs. 39%, PCV) in the *L*-citrulline and control groups 2 h before the race; as well as the concentrations of the PO_2_ (46.25 vs. 44.47 mmHg) and Hct (56.5 vs. 57.67%, PCV) immediately after the race; and the concentrations of the PO_2_ (35.53 vs. 33.08 mmHg) and Hct (36.83 vs. 38.33%, PCV) 2 h after the race all exhibited a trend of initially increasing and then decreasing throughout the entire period. However, there were no significant differences between the *L*-citrulline and control groups (*p* > 0.05).

### 3.3. Effects of L-citrulline Supplementation on Hematologic Parameters and Plasma Biochemistry of Yili Horses before and after 2000 m Speed Race

As shown in Figure 4, the number of monocytes in the plasma of the horses in the *L*-citrulline group was 27.54% lower than that of the control group 2 h before the race (*p* < 0.05), and the rest of the hematologic parameters did not change significantly (*p* > 0.05). At 2 h before the race, the total protein content of the *L*-citrulline group was significantly higher than that of the control group, by 6.29% (*p* < 0.05), and there was a trend towards an increase in the albumin content (*p* = 0.067). At 2 h after the race, the total protein content of the *L*-citrulline group was significantly higher than that of the control group, by 3.93% (*p* < 0.05), and there was a trend towards an increase in the globulin content (*p* = 0.084). In the immediate post-competition period, the plasma creatinine levels were significantly higher in the *L*-citrulline group than in the control group, by 13.91% (*p* < 0.05). Compared to 2 h before the race, the plasma creatine kinase and lactate dehydrogenase activities significantly increased immediately after the competition in both the *L*-citrulline and control groups. Moreover, the lactate dehydrogenase activities in the *L*-citrulline group were significantly higher than those in the control group at 2 h before and 2 h after the race (*p* < 0.05).

### 3.4. Effects of L-citrulline Supplementation on Antioxidant Capacity and NO Concentration of Yili Horses before and after 2000 m Speed Race

As shown in Figure 5, at 2 h before the race, the SOD, CAT, and NO contents of the *L*-citrulline group were significantly higher than those of the control group, by 25.35% (*p* < 0.05), 14.62% (*p* < 0.05), and 16.44% (*p* > 0.05), respectively, and the T-AOC content of the *L*-citrulline group was significantly higher than that of the control group, by 30.68% (*p* < 0.01). In the immediate post-competition period, T-AOC levels were significantly higher in the *L*-citrulline group than in the control group, by 9.13% (*p* < 0.05), and there was a trend towards higher plasma NO concentrations (*p* = 0.082). At 2 h after the race, the concentrations of SOD, CAT, T-AOC, and NO in the *L*-citrulline group were higher than those in the control group, by 37.03% (*p* < 0.05), 10.65% (*p* < 0.05), 24.21% (*p* < 0.01), and 13.03% (*p* > 0.05), respectively, while the concentration of MDA was lower than that in the control group, by 11.92% (*p* < 0.05).

### 3.5. Effects of L-citrulline Supplementation on Metabolism Concentrations of Plasma Amino Acids in Yili Horses

In Table 7 and Figure 6A, the concentrations of citrulline and arginine in the *L*-citrulline group were significantly higher than those of the control group, at 25.25% (*p* < 0.05) and 26.88% (*p* < 0.05), respectively. In terms of essential amino acids, the concentration of lysine in the *L*-citrulline group was higher than that of the control group, by 26.21%, but it was not statistically significant (*p* > 0.05), while the concentration of threonine in the *L*-citrulline group was significantly lower than that of the control group, by 26.61% (*p* < 0.05). In terms of non-essential amino acids, the concentrations of alanine, glycine, serine, and histidine in the *L*-citrulline group were lower than those in the control group, by 17.79% (*p* < 0.05), 29.18% (*p* < 0.01), 18.46% (*p* < 0.05), and 13.96% (*p* > 0.05), respectively, and the concentration of glutamine in the *L*-citrulline group was higher than that in the control group, by 10.07% (*p* < 0.05). There were no significant differences in the concentrations of the total amino acids, essential amino acids, and non-essential amino acids between the groups.

As shown in Figure 6B, the principal component analysis (PCA) indicates that the horizontal and vertical axes represent two principal components, with each point corresponding to a sample. The horizontal axis represents the primary factor influencing sample variance, accounting for 35% of the total variance. Subsequently, the vertical axis represents the second principal component, contributing 15% to the sample variance. Together, these two principal components account for 50% of the cumulative variance, indicating that *L*-citrulline shows a significant similarity in the composition of amino acids between the *L*-citrulline and control groups. To explore the relationship between *L*-citrulline and free amino acids, a Pearson correlation analysis was performed on the *L*-citrulline group and amino acids, followed by the generation of a heatmap (Figure 6C). Positive correlations between the amino acids are indicated in red, while negative correlations are shown in blue. A correlation coefficient with an absolute value greater than 0.8 indicates a significant correlation (*p* < 0.05), and one greater than 0.9 indicates a highly significant correlation (*p* < 0.01). As shown in this Figure, citrulline is highly significantly positively correlated with ornithine, glutamic acid, and aspartic acid and significantly positively correlated with arginine, phenylalanine, and methionine. Arginine is significantly positively correlated with alanine and ornithine. Ornithine is highly significantly positively correlated with aspartic acid and significantly positively correlated with arginine, glutamic acid, phenylalanine, methionine, asparagine, and cysteine. Glutamic acid is highly significantly positively correlated with aspartic acid and ornithine and significantly positively correlated with phenylalanine and methionine. Phenylalanine is highly significantly positively correlated with aspartic acid and significantly positively correlated with glutamic acid and methionine. The results suggest that *L*-citrulline treatment may lead to a synergistic enhancement in arginine, ornithine, glutamic acid, and phenylalanine, potentially affecting the levels of certain free amino acids.

### 3.6. Effects of L-citrulline Supplementation on Plasma Polyamine Concentrations in Yili Horses

As shown in Table 8, the concentrations of putrescine and spermidine in the *L*-citrulline group were significantly higher than those in the control group, by 41.08% (*p* < 0.05) and 40.71% (*p* < 0.05), respectively, and the concentration of spermine in the *L*-citrulline group was higher than that in the control group, by 12.14%, but this difference was not significant (*p* > 0.05).

## 4. Discussion

In our study, at the experiment’s outset, the 2000 m exercise performance for both groups of speed-racing *Yili* horses was recorded at 219.80 s and 217.20 s, respectively. After 38 days, the group of speed-racing *Yili* horses supplemented with *L-*citrulline exhibited a decrease in their 2000 m exercise performance, by 5.40 s, compared with the control group, in terms of race time. This indicates that the 2000 m race time was reduced by 2.51% compared to the control group and by 3.31% compared to the initial trial. In competitive sports, even a marginal advantage in seconds can significantly impact the final ranking [17]. Suzuki et al. (2016) observed that professional cyclists significantly reduced their 4 km race completion time by 1.5% after *L-*citrulline supplementation for 7 days [18]. We hypothesize that the potential mechanism behind this improvement may be the increased plasma arginine concentration due to *L-*citrulline supplementation. Elevated arginine levels can enhance the bioavailability of nitric oxide synthesis substrates. Nitric oxide, by binding to cytochrome c oxidase in mitochondria, improves mitochondrial respiration regulation, thereby limiting oxygen consumption and promoting a better oxygen distribution within skeletal muscles [19]. This, in turn, improves the matching of muscle oxygenation with metabolic demands. In this experiment, the supplementation of *L*-citrulline did not significantly improve the race performance in horses. One reason could be that citrulline may require long-term continuous supplementation. Additionally, citrulline enhances the cellular oxidative phosphorylation system, improving oxygen utilization, which is more suitable for endurance sports. Short-term, high-intensity citrulline supplementation may not be sufficient to significantly enhance athletic performance.

Abnormal changes in respiration and heart rates can severely impact the aerobic metabolic capacity of sport horses, ultimately leading to a decrease in the racing speed and the onset of exercise-induced fatigue symptoms. In this experiment, the *L*-citrulline group of horses showed little variation in respiration and heart rates compared to the control group 2 h before the race. Immediately after the race, the horses exhibited markedly increased breathing and heart rates, yet there remained no significant difference between the *L*-citrulline group and the control group. During the 2 h post-race recovery phase, the *L*-citrulline group showed a slight trend of a decreased heart-rate frequency. In summary, this indicates that *L*-citrulline not only enhances the race speed in horses but also plays a role in facilitating the rapid recovery of physiological functions post-exercise.

During high-intensity exercises in horses, as the rate of glycolysis increases, there is a substantial accumulation of H+, resulting in increased lactate levels in the body. This contributes to a decrease in the muscle pH and an increase in fatigue sensations, causing alterations in the blood acid–base balance capacity [20]. In our study, there were no significant differences in the pH, TCO_2_, PCO_2_, Na^+^, K^+^, Ca^2+^, and Lac between the two groups 2 h before the race. However, immediately after the race, the blood lactate levels in the *Yili* horses supplemented with *L-*citrulline were significantly lower than those in the control group. Similar results have also been described by Kiyici [21] and Takeda [22]. We hypothesize that during high-intensity exercises, supplementation with *L*-citrulline may inhibit the increase in lactate levels, aiding in the aerobic utilization of pyruvate within cells, thereby reducing lactate production through anaerobic pathways. At the onset of an exercise, the ATP required by the muscles is initially supplied through aerobic metabolism. As the exercise continues and the demand for ATP increases beyond the capacity of aerobic metabolism, glycolysis becomes the primary pathway to meet the short-term ATP requirements. Additionally, intense exercises may cause hypoxemia in horses, primarily due to reduced oxygen diffusion to the pulmonary capillaries during exercises [23]. Consequently, the mismatch between the oxygen supply and demand can impair the exercise performance of horses. Studies have shown that *L-*citrulline can improve the availability of muscle oxygen, increase the rate of pulmonary oxygen uptake, and enhance one’s tolerance to a high-intensity exercise [24]. In our experiment, it was found that the overall improvement in exercise performance after *L-*citrulline supplementation might be attributed to the metabolism of *L-*citrulline into NO, which promotes vasodilation by reducing cellular calcium levels. This leads to an increased blood flow in the muscles, thereby enhancing the PO_2_ content transported in the body’s blood. Additionally, NO can reduce oxygen consumption during an exercise, improve muscle contractility, and decrease ATP consumption. However, there was no significant difference in the PO_2_ levels between the two groups, indicating that the horses may not have experienced symptoms of hypoxemia. Nevertheless, the accurate determination of the extent of hypoxia in horses requires an analysis of the arterial blood gases. During exercises, a higher concentration of hemoglobin (Hb) is crucial for maintaining an acid–base balance and enhancing athletic performance. Oxygen delivery is limited by the oxygen-carrying capacity of hemoglobin, which also serves as a buffer for the lactic acid produced during muscle contractions. Blood glucose (Glu) primarily provides energy during exercises. Higher hematocrit (Hct) levels lead to increased blood viscosity, which in turn raises the blood flow resistance. An elevated blood viscosity can impair the function of numerous bodily systems and organs. Conversely, a lower blood viscosity can enhance oxygen transport in the blood, thus promoting performance during the recovery period after an exercise [25]. In our study, the Glu levels in the *L*-citrulline group were higher than those in the control group, while the Hct concentrations were lower, though these differences were not statistically significant. Immediately post-competition, the Hb content was significantly higher in the *L*-citrulline group compared to the control group, likely due to the increased metabolic demands during the intense exercise. *L-*citrulline helps enhance energy metabolism efficiency, optimizes oxygen uptake and utilization, and indirectly stimulates the production or release of Hb to meet the higher oxygen demands. Under the stimulus of a short-term high-intensity exercise, the homeostasis of horses’ bodies is disrupted, leading to changes in blood gas indicators. In our study, immediately post-competition, the levels of Hb, Glu, Hct, Lac, and PO_2_ significantly increased compared to the pre-competition levels, while TCO_2_ and PCO_2_ significantly decreased. At 2 h post-competition, these levels gradually returned to their pre-competition values. In summary, the aforementioned indicators in the venous blood of horses before and after exercises are interconnected and mutually regulated. Although no significant changes occurred between the *L*-citrulline group and the control group, both groups returned to normal ranges 2 h post-race, which indicates that the blood gas transport capacity was within normal limits under the exercise intensity of this 2000 m simulated race for 2-year-old *Yili* horses.

In this study, the number of monocytes in the *L*-citrulline group was significantly higher than that in the control group 2 h before the race. The monocyte system plays a crucial role in the body’s immune defense, inflammatory response, tissue repair, and metabolic regulation. This might be due to citrulline affecting the immune system through NO, which can inhibit the adhesion and migration of monocytes, thereby influencing their aggregation at inflammation sites [26]. William et al. (2012) found that the addition of *L-*citrulline can improve conditions in patients with sickle cell disease, maintaining the total number of leukocytes and neutrophils at near-normal levels [27]. This study found that, aside from a significant change in the number of monocytes, no other hematologic indices showed statistically significant differences. This indicates that adding 50 g of *L-*citrulline in a training supplementation regimen for at least 38 days is safe for athletic horses. The final product of protein metabolism in the body is urea. *L-*citrulline is a precursor for the synthesis of L-arginine, which, through the urea cycle in the liver, removes excess ammonia by converting it into urea, playing a crucial role in regulating nitrogen homeostasis [28]. Wu et al. (2018) found that the oral administration of 8 g of *L-*citrulline in adult sheep had no significant effect on urea concentrations, which is consistent with the results of this study [29]. Fligge (1997) demonstrated that the addition of 500 mg/kg of arginine to the diet of calves significantly increased plasma urea levels [30]. This finding is inconsistent with the results of the present study. Although there was an increase in the urea levels in this study, it did not reach a statistically significant difference. This discrepancy may be due to a reduced activity of arginase or the breakdown and utilization of L-arginine by various enzymes in the body. In this experiment, the total protein content of the *L*-citrulline group was significantly higher than that of the control group at 2 h before and 2 h after the race, and there was a non-significant increase in the levels of albumin and globulin in the experimental group compared with the control group at 2 h before and 2 h after the race, respectively. These findings are consistent with Li’s (2021) study, which demonstrated that the supplementation of *L-*citrulline in Thoroughbreds aged 0–3 months significantly increased albumin levels [31]. Similarly, Sun’s (2012) experiment showed that *L-*citrulline supplementation in suckling piglets significantly increased the levels of total protein and globulin, indicating that *L-*citrulline can improve protein metabolism and has a positive effect on the nutritional status of the body [32]. In this experiment, the Cre content in the *L*-citrulline group immediately after the race was significantly higher than that in the control group, possibly due to an increase in L-arginine leading to an increase in creatine, as *L*-arginine is used for Cre synthesis [33]. In our study, CK and LDH activities sharply increased immediately after the race, indicating that the horses’ muscles might have been damaged during the 2000 m exercise, with an increased oxidation and decomposition of amino acids in the muscles for energy supply. Although there was no significant difference in the CK activity between the two groups immediately after the race, the *L*-citrulline group had lower CK activity compared to the control group, suggesting that *L-*citrulline may influence the degree of muscle damage. Under hypoxic conditions, lactate dehydrogenase (LDH) catalyzes the conversion of pyruvate to lactate while oxidizing NADH (reduced form of coenzyme I) to NAD+ (oxidized form of coenzyme I), facilitating energy production via glycolysis. Under normal aerobic metabolic conditions, LDH is involved in converting lactate to pyruvate, allowing it to enter the mitochondria for aerobic oxidation, thereby generating more energy [34]. In this experiment, the activity of LDH was significantly higher at 2 h before and after the competition compared to the control group. This finding aligns with the study by Martínez-Sánchez (2017), where male runners who consumed 500 mL of watermelon juice containing 3.45 g of *L-*citrulline showed a significant decrease in plasma lactate levels and a significant increase in lactate dehydrogenase concentrations after a half marathon [35]. This suggests that *L-*citrulline supplementation may enhance the production of aerobic energy and promote the increased activity of this enzyme, aiding in the post-exercise recovery process. Therefore, the observed changes in these indicators demonstrate that exercise-induced oxidative stress can indeed impact markers related to exercise-induced injuries in the body.

Reactive oxygen species (ROS) are markers of normal cellular respiration [36]. When ROS accumulate excessively, they lead to oxidative stress and disrupt the balance of the body’s endogenous antioxidant defenses, resulting in an increased cellular inflammatory response. During high-intensity exercises in horses, the body’s oxygen utilization increases, which in turn causes a continuous production of reactive oxygen species and free radicals from cells. Studies have shown that muscle damage and oxidative stress are significantly increased after exercises [37]. This leads to an elevated metabolic oxidation process accompanied by ROS production and tissue damage. When the free radicals generated by cells exceed the physiological clearance capacity, cellular damage occurs. Therefore, the addition of exogenous antioxidants is required to regulate the redox balance in horses [38]. *L-*citrulline can enhance the production of endothelial nitric oxide synthase (eNOS) in endothelial cells. eNOS catalyzes the production of NO, which can rapidly react with superoxide anion radicals to form relatively stable peroxynitrite (ONOO-), thereby reducing the concentration of superoxide anion radicals and mitigating their oxidative damage to cells [39]. However, *L-*citrulline can also exert antioxidant functions independently of NO by directly reducing the formation of hydroxyl radicals [40]. In this experiment, it was found that 2 h before the competition, the *L*-citrulline group showed a significant increase in SOD and CAT activities and a highly significant increase in T-AOC levels. This indicates that with the accumulation of the trial duration, *L-*citrulline itself has the ability to effectively enhance the antioxidant capacity. Valaei et al. (2021) demonstrated that *L-*citrulline supplementation one hour before an exercise can be an effective antioxidant, significantly increasing post-exercise SOD, GSH-px, and CAT activities [41]. This is similar to the results of our experiment, which showed a significant increase in the T-AOC levels immediately after the competition. This indicates that *L-*citrulline can directly enhance the body’s antioxidant capacity. Evidence suggests that nutritional supplements containing antioxidant substances can promote recovery by reducing exercise-induced oxidative stress, thereby positively affecting athletic performance [42]. At 2 h post-competition, *L-*citrulline significantly increased the activities of SOD and CAT while significantly reducing the MDA levels and markedly increasing the T-AOC levels. This indicates that *L-*citrulline supplementation can enhance the plasma antioxidant capacity of *Yili* horses, aiding in the elimination of free radicals and reducing exercise-induced damage to promote recovery. This effect may be attributed to the rapid metabolism of *L-*citrulline into arginine and NO in the *Yili* horses, which indirectly increases the activity of related antioxidant enzymes. Our study results indicate that *L*-citrulline supplementation showed a trend towards increased plasma NO concentrations immediately after the race. This is consistent with the findings of Zhang et al. (2019), who reported that a 60 g arginine supplementation significantly increased plasma NO levels and significantly reduced plasma ammonia levels in *Yili* horses [43]. Similarly, Liu et al. (2021) demonstrated that *L-*citrulline supplementation significantly increased plasma NO levels [44]. This effect may be due to *L-*citrulline supplementation increasing the plasma arginine concentration, as arginine is a substrate for nitric oxide synthase (NOS). An NOS enzyme catalyzes a complex enzymatic reaction, leading to the formation of NO from arginine and molecular oxygen [45], thereby increasing the NO content in the plasma. The production of NO influences muscle function by inducing smooth muscle relaxation and vasodilation through the NO–cGMP pathway [46]. While *L-*citrulline supplementation has a positive effect on exercise performance, this impact is not significant. This may be due to variations in exercise protocols, the amount of supplementation, and the duration of supplementation.

Amino acids are vital nutrients within horses, essential for protein synthesis, energy supply, and disease resistance during various physiological processes. Studying the combination of exercise training and *L-*citrulline supplementation can provide a deeper understanding of amino acid changes in *Yili* speed-racing horses. *L-*citrulline is a non-protein amino acid that exhibits a highly specific metabolism. It is primarily metabolized in the kidneys, where it is converted into arginine by a proximal tubule cell [47]. This conversion in the kidneys occurs through the partial urea cycle involving argininosuccinate synthase and argininosuccinate lyase [48], and the synthesized arginine is released into the systemic circulation. Gilbreath et al. (2020) reported that supplementing the diet with 8 g/day of *L-*citrulline in adult sheep significantly increased plasma citrulline concentrations and significantly raised arginine concentrations four hours after feeding [49]. Ma et al. (2023) studied the effects of supplementing 8-month-old Hu sheep ewes with *L-*citrulline and found that plasma citrulline concentrations significantly increased, arginine concentrations increased, and ornithine concentrations did not show significant changes 4 h post-supplementation [50]. Similarly, Gilbreath et al. (2020) demonstrated that adding 0.25% citrulline to the diet of mature steers resulted in increased plasma concentrations of citrulline and arginine 4 h after feeding [51]. Daniel et al. found that the oral administration of *L-*citrulline can increase plasma concentrations of *L-*citrulline and arginine, with dose and time effects [12]. Therefore, considering the overall experimental design, measuring the plasma amino acid concentrations on the 20th day of the experiment, specifically 3 h after supplemental feeding, may more accurately reflect the metabolic interactions between amino acids following citrulline supplementation. In this experiment, it was found that supplementing speed-racing *Yili* horses with *L*-citrulline resulted in significantly higher concentrations of citrulline and arginine in the plasma compared to the control group, while the ornithine concentrations did not show significant changes. This indicates that *L-*citrulline is absorbed by intestinal cells in the portal circulation, bypasses the metabolism of periportal hepatocytes, and is transported to the kidneys. Approximately 80% of it is metabolized into arginine by argininosuccinate synthetase and argininosuccinate lyase and then released into the bloodstream for systemic use. Studies have shown that arginine is degraded into ornithine by the action of arginase, and *L-*citrulline may inhibit the activity of arginase, acting as a potent allosteric inhibitor [52] This could result in a decreased concentration of ornithine. It can be concluded that the exogenous addition of *L*-citrulline influences the concentrations of citrulline and arginine in the plasma of exercising horses. It may also affect ornithine concentrations, though this effect was not significant in this experiment. In both animal and human skeletal muscles, the availability of intracellular amino acids can be regulated through the mechanistic target of the rapamycin complex 1 (mTORC1) signaling pathway [53]. Although *L-*citrulline is not a component of proteins, it can stimulate protein synthesis in skeletal muscles via the mTOR signaling pathway mechanism. Therefore, *L-*citrulline plays a crucial role in maintaining protein homeostasis [54]. In this experiment, the glutamine concentration in the *L*-citrulline group was significantly higher than that in the control group, while the levels of alanine, glycine, serine, and threonine were significantly lower. This suggests that the combination of *L-*citrulline supplementation and exercise training may affect the absorption and utilization of certain amino acids in *Yili* horses used for speed racing. Skeletal muscle can synthesize glutamine from glutamate, ATP, and ammonia through the cytoplasmic glutamine synthetase reaction, which may lead to an increase in glutamine concentrations. The intracellular concentration of amino acids can be regulated through the coupled transport process of the System N-neutral amino acid transporter 2 (SNAT2) [55]. The alanine transported by SNAT2 is a significant participant in the glucose–alanine cycle between the muscle and liver. In this cycle, the ammonia in the muscle is transported to the liver in the non-toxic form of alanine. Alanine undergoes deamination to produce pyruvate, which is then used in gluconeogenesis to generate the glucose needed by the muscle. Supplementation with *L-*citrulline and exercise training may increase the utilization rate of pyruvate in the muscle, leading to a decrease in alanine concentrations. The decrease in the concentrations of glycine, serine, and threonine may be related to the tissue metabolism and absorption of free amino acids from the plasma. Sureda et al. (2009) demonstrated that *L-*citrulline supplementation can enhance the overall utilization of nitrogen, thereby promoting protein synthesis and increasing the protein content in muscles during exercises [56], which in turn promotes the utilization of amino acids. Therefore, adding *L-*citrulline to the diet provides a reference for understanding the changes in the content of free amino acids in exercising horses.

Citrulline can stimulate the expression of ornithine decarboxylase (ODC), and under the influence of ODC, ornithine promotes the synthesis of polyamines (putrescine, spermidine, and spermine). Evidence suggests that maintaining normal polyamine levels is crucial for cellular functions. Polyamines have been observed to significantly impact various key cellular functions, including the regulation of cell proliferation and differentiation, protein synthesis, exercise, muscle development, antioxidant activity, and other stress responses [57]. Additionally, high levels of spermine in the mitochondrial matrix can regulate Ca^2+^ entry into mitochondria, which may influence the activity of the pyruvate dehydrogenase complex or directly affect mitochondrial functions [58]. Li X et al. (2023) demonstrated that the addition of citrulline to a diet significantly increased ornithine decarboxylase activity, NO, and polyamine synthesis [59]. In this study, supplementation with *L-*citrulline significantly elevated the concentrations of putrescine and spermidine, with the spermine levels also being higher than those in the control group. This effect may be due to citrulline enhancing the expression of ornithine decarboxylase. Therefore, *L-*citrulline supplementation can increase the plasma polyamine metabolism concentration in *Yili* horses. As cytokines, polyamines are involved in regulating inflammatory responses, promoting the repair of damaged tissues and inducing the production of anti-inflammatory proteins. They are also essential for the activation of the eukaryotic translation initiation factor 5A (EIF5A). The regulation of gene transcriptions can affect mitochondrial functions in macrophages; a decrease in EIF5A activity reduces the mitochondrial oxygen demand, leading to the differentiation of macrophages towards a pro-inflammatory phenotype [60]. The experiment by Luchessi (2009) with mice confirmed the hypothesis that EIF5A is involved in protein synthesis, a process that requires a substantial supply of ATP [61]. Therefore, the influence of EIF5A on this process may be related to the regulation of cellular energy metabolism. A reduced ODC activity and decreased intracellular polyamine levels can induce a significant increase in the number of pro-inflammatory macrophages, leading to increased inflammation in the stomach and colon of animals.

## 5. Conclusions

In conclusion, supplementation with *L*-citrulline can affect the circulating concentrations of *L*-citrulline and arginine in *Yili* horses. In the immediate post-race period, it can enhance aerobic pathways, reduce lactate levels, and increase hemoglobin concentrations, thereby improving the recovery of the acid–base balance post-exercise. Furthermore, this supplementation can enhance the antioxidant capacity and regulate physiological and biochemical levels, providing valuable insights into the exercise performance and post-exercise recovery of athletic horses.

## Figures and Tables

**Figure 1 animals-14-02438-f001:**
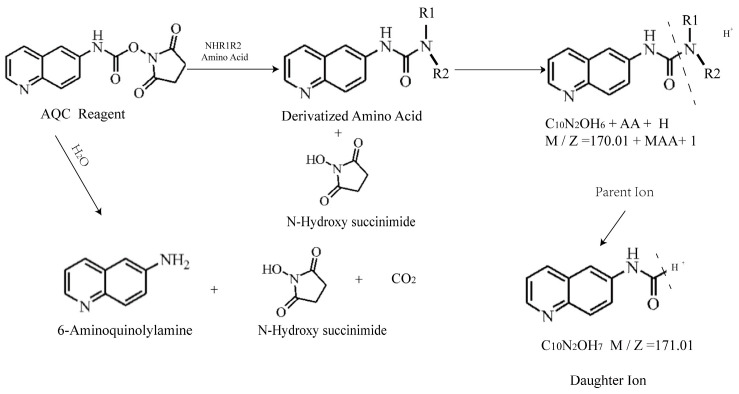
Principle of amino acid derivatization.

**Figure 2 animals-14-02438-f002:**
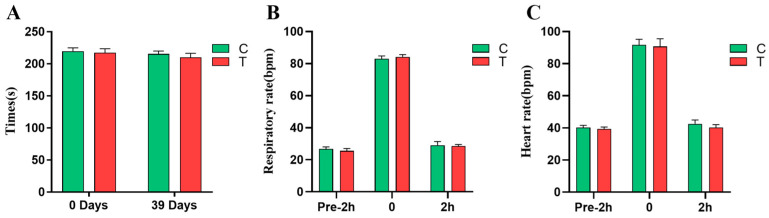
Effects of *L*-citrulline supplementation on race time, respiration, and heart rate of *Yili* horses before and after the 2000 m speed race. (**A**–**C**) (C, control group; T, *L*-citrulline group). Each bar represents the mean of 6 horses ± the standard deviation of the mean (SD).

**Figure 3 animals-14-02438-f003:**
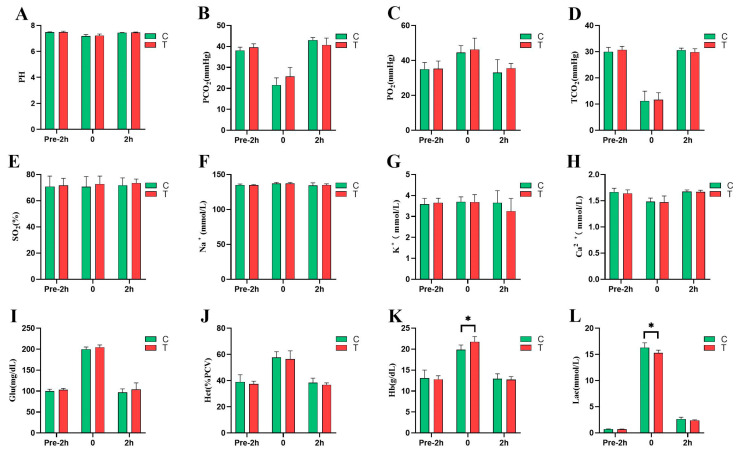
Effects of *L*-citrulline supplementation on blood gas indices of *Yili* horses before and after the 2000 m speed race. (**A**) pH, blood pH; (**B**) PCO_2_, partial pressure of carbon dioxide; (**C**) PO_2_, partial pressure of oxygen; (**D**) TCO_2_, partial pressure of carbon dioxide; (**E**) SO_2_, oxygen saturation of hemoglobin; (**F**) Na^+^, ionized sodium; (**G**) K^+^, ionized potassium; (**H**) Ca^2+^, ionized calcium; (**I**) Glu, blood glucose; (**J**) Hct, hematocrit; (**K**) Hb, hemoglobin; (**L**) Lac, lactate (C, control group; T, *L*-citrulline group). Each bar represents the mean of 6 horses ± the standard deviation of the mean (SD). * indicates a significant difference between the two groups (*p* < 0.05).

**Figure 4 animals-14-02438-f004:**
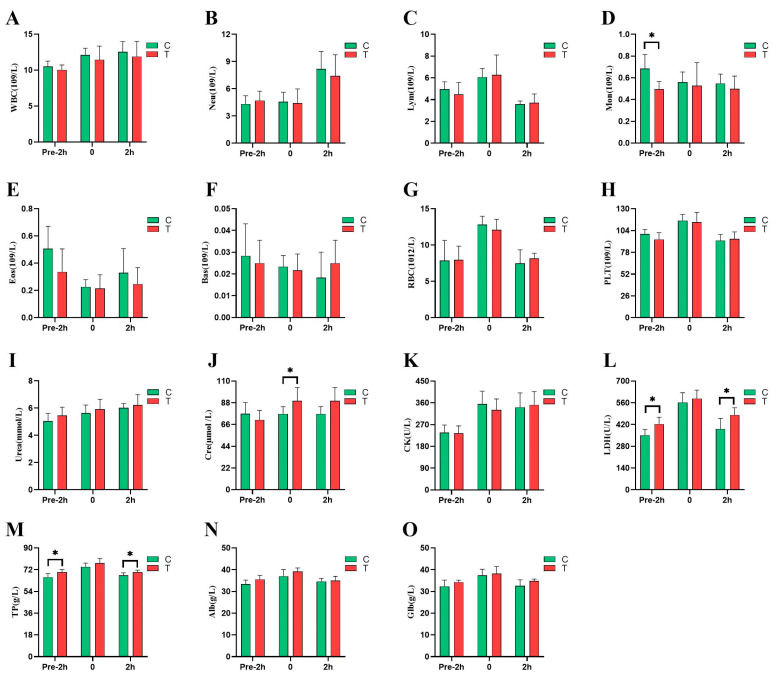
Effects of *L*-citrulline supplementation on hematologic parameters and plasma biochemistry of *Yili* horses before and after the 2000 m speed race. (**A**) WBCs, white blood cells; (**B**) Neus, neutrophils; (**C**) Lyms, lymphocytes; (**D**) Mons, monocytes; (**E**) Eos, eosinophils; (**F**) Bas, basophils; (**G**) RBCs, red blood cells; (**H**) PLTs, platelets; (**I**) Urea, urea; (**J**) Cre, creatinine; (**K**) CK, creatine kinase; (**L**) LDH, lactate dehydrogenase; (**M**) TP, total protein; (**N**) Alb, albumin; (**O**) Glb, globulin. (C, control group; T, *L*-citrulline group). Each bar represents the mean of 6 horses ± the standard deviation of the mean (SD). * indicates a significant difference between the two groups (*p* < 0.05).

**Figure 5 animals-14-02438-f005:**
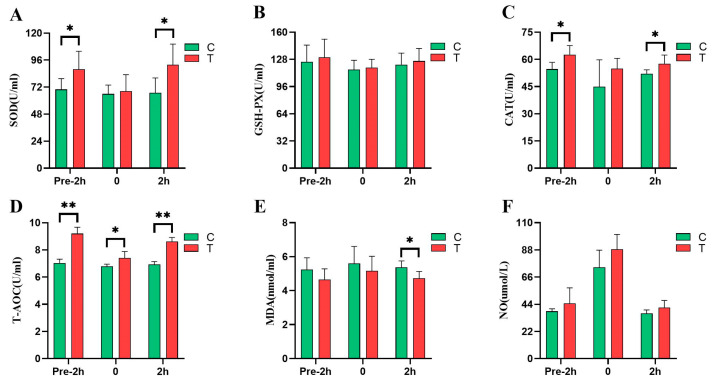
Effects of *L*-citrulline supplementation on antioxidant capacity and NO concentration of *Yili* horses before and after 2000 m speed race. (**A**) SOD, superoxide dismutase; (**B**) GSH-PX, glutathione peroxidase; (**C**) CAT, catalase; (**D**) T-AOC, total antioxidant capacity; (**E**) MDA, malondialdehyde; and (**F**) NO, nitric oxide. (C, control group; T, *L*-citrulline group). Each bar represents the mean of 6 horses ± the standard deviation of the mean (SD). * indicates a significant difference between the two groups (*p* < 0.05), and ** indicates a highly significant difference between the two groups (*p* < 0.01).

**Figure 6 animals-14-02438-f006:**
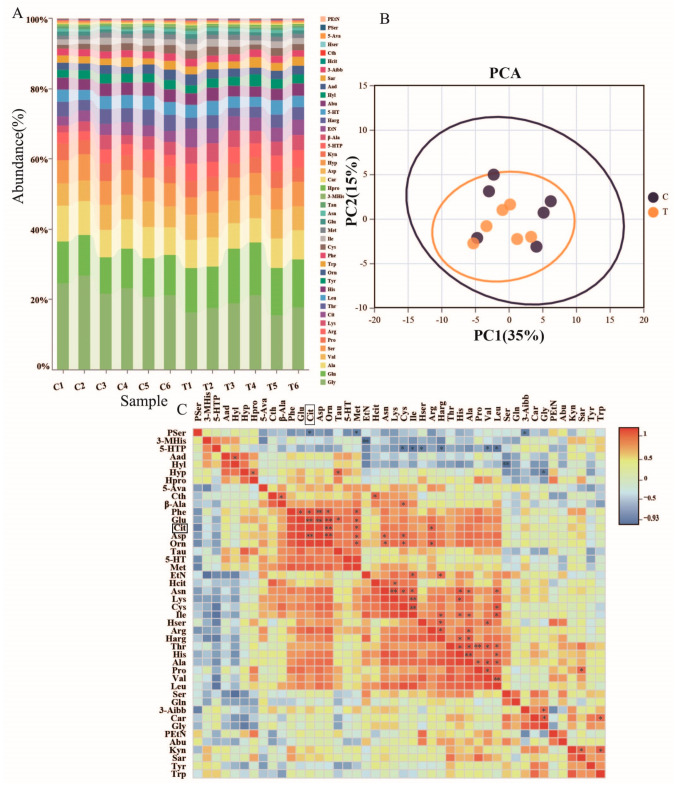
Effects of the supplemental feeding of *L*-citrulline on amino acid metabolism in the plasma of speed-racing Yili horses. (**A**) Stacked distribution plot of amino acids, (**B**) PCA plot of amino acids, (**C**) correlation analysis of amino acids in the *L*-citrulline group. (C, control group; T, *L*-citrulline group). * indicates a significant difference between the two groups (*p* < 0.05), ** indicates a highly significant difference between the two groups (*p* < 0.01).

**Table 1 animals-14-02438-t001:** Composition and nutritional levels of the concentrate supplement (DM basis, %).

Items	Content
Concentrate supplement composition	
Corn	50.4
Wheat bran	17.2
Soybean meal	27.14
Ca(H_2_PO_4_)_2_	3.4
NaCl	0.86
Premix ^(1)^	1.00
Total	100.00
**Nutritional level** ^(2)^	
DM	89.25
CP	14.96
EE	6.02
NDF	13.82
ADF	9.29
Ash	8.27
Ca	0.57
P	0.23

^(1)^ The premix provided the following per kg of the concentrate supplement: Vit A, 15 mg; Vit B_1_, 21.28 mg; Vit B_2_, 333.1 mg; Vit B_6_, 1.23 mg; Vit D, 2.3 mg; Vit E, 850 mg; pantothenic acid, 4.66 mg; biotin, 5 mg; nicotinamide, 12.13 mg; Cu (as copper sulfate), 42.02 mg; Fe (as ferrous sulfate), 111.93 mg; Mn (as manganese sulfate), 185.86 mg; Zn (as zinc sulfate), 176.03 mg; I (as potassium iodide), 29.52 mg; Se (as sodium selenite), 42.28 mg; Co (as cobalt chloride), 4.15 mg. ^(2)^ Nutritional levels was measured.

**Table 2 animals-14-02438-t002:** Nutritional levels of the dried forage (DM basis, %).

Nutritional Level	
DM	90.57
CP	10.04
EE	2.54
NDF	56.50
ADF	30.57
Ash	9.35
Ca	0.77
P	0.08

Note: nutritional levels were measured.

**Table 3 animals-14-02438-t003:** Chromatographic and mass spectrometry conditions of amino acids.

Chromatographic column: Venusil MP C18 (150 × 4.6 mm, 5 µm)	Column temperature: 50 °C
Mobile phase A: acetonitrile (0.1% formic acid + 0.01% heptafluorobutyric acid	Mobile phase B: water (0.1% formic acid + 0.01% heptafluorobutyric acid)
Flow rate: 1 mL/min	Injection volume: 3 µL
Ion source: + ESI electrospray ion source	Scanning method: MRM multi-reaction monitoring
Curtain gas: 20 psi	Spray voltage: +5500 V
Collision gas: medium	Collision chamber ejection voltage: 2.0 V
Nebulizer: 55 psi	Auxiliary gas: 60 psi
Nebulization temperature: 500 °C	Ingress voltage: 10 V

**Table 4 animals-14-02438-t004:** Standard curves and peak times of amino acids.

Items	Molecular Weight	Peak Time (Minutes)	Standard Curve	R
3-MHis	169.18	3.86	y = 9.34 × 10^−5^x + 1.76 × 10^−5^	0.9960
3-Aib	103.12	8.73	y = 0.00935x + −0.000623	0.9997
5-Ava	117.15	9.54	y = 0.0142x + −0.000157	0.9999
5-HT	176.20	9.46	y = 0.00243x + −0.003	0.9916
5-HTP	220.22	8.97	y = 0.00147x + −0.00342	0.9965
Aad	161.16	6.37	y = 0.00205x + −5.77 × 10^−5^	0.9998
Abu	103.12	8.73	y = 0.00863x + −0.00142	0.9990
Ala	89.09	6.49	y = 0.00286x + 0.0245	0.9948
Arg	174.20	5.18	y = 0.000617x + −0.00114	0.9932
Asn	132.12	3.28	y = 0.00467x + 0.000919	0.9982
Asp	133.10	4.56	y = 0.000931x + −0.000239	0.9980
β-Ala	89.09	6.49	y = 0.0127x + 0.00262	0.9991
Car	226.23	4.64	y = 0.000166x + −5.19 × 10^−5^	0.9962
Cit	175.19	4.42	y = 0.000381x + −0.000156	0.9976
Cth	222.26	8.64	y = 0.000223x + −2.41 × 10^−5^	0.9981
Cys	121.16	8.94	y = 4.87 × 10^−5^x + −6.69 × 10^−6^	0.9992
EtN	61.08	4.49	y = 0.00123 x + −0.000252	0.9975
Gln	146.15	3.53	y = 0.00144x + −0.00577	0.9981
Glu	147.13	4.84	y = 0.00254x + 0.000202	0.9972
Gly	75.07	4.62	y = 0.000886x + 0.000291	0.9972
Harg	188.20	6.31	y = 0.000479x + −0.000147	0.9952
Hcit	189.20	5.58	y = 0.0005x + −0.000157	0.9971
His	155.15	3.85	y = 0.000168x + −8.75 × 10^−5^	0.9929
Hser	119.10	5.31	y = 0.00216x + −0.000496	0.9972
Hyl	162.20	9.00	y = 0.000167x + −5.72 × 10^−5^	0.9968
Hyp	131.13	3.13	y = 0.0318x + 0.0185	0.9923
lle	131.18	11.26	y = 0.0106x + 0.00278	0.9988
Kyn	208.10	10.62	y = 0.00155x + −0.000359	0.9972
Leu	131.17	11.26	y = 0.00939x + 0.00834	0.9980
Lys	146.17	8.93	y = 0.000523x + −0.000668	0.9947
Met	149.20	9.28	y = 0.00438x + 0.0018	0.9984
Orn	132.16	8.87	y = 0.000701x + −0.00024	0.9972
Hpro	129.16	10.01	y = 0.000924x + −6.7 × 10^−6^	0.9993
PEtN	141.06	1.94	y = 0.00444x + −0.000254	0.9982
Phe	165.19	11.37	y = 0.00633x + 3.5 × 10^−5^	0.9983
Pro	115.14	7.39	y = 0.00571x + 0.0172	0.9946
PSer	185.07	1.75	y = 0.000173x + −1.02 × 10^−5^	0.9942
Sar	89.09	6.49	y = 0.00624x + −0.00136	0.9977
Ser	105.09	3.72	y = 0.00341x + 0.00317	0.9967
Tau	125.25	2.06	y = 0.00154x + 0.0128	0.9920
Thr	119.13	5.31	y = 0.0029x + −0.00207	0.9985
Trp	204.23	11.60	y = 0.00383x + −0.00102	0.9965
Tyr	181.19	9.02	y = 0.00287x + 0.00249	0.9969
Val	117.15	9.54	y = 0.00519x + 0.119	0.9941

**Table 5 animals-14-02438-t005:** Chromatographic and mass spectrometry conditions of polyamines.

Chromatographic column: Agela Venusil MP C18 (100 × 4.6 mm, 3 µm)	Column temperature: 50 °C
Mobile phase A: acetonitrile (0.1% formic acid)	Mobile phase B: water (0.1% formic acid)
Flow rate: 1 mL/min	Injection volume: 10 µL
Ion source: + ESI electrospray ion source	Scanning method: MRM multi-reaction monitoring
Curtain gas: 20 psi	Spray voltage: +5000 V
Collision gas: medium	Collision chamber ejection voltage: +2.0 V
Nebulizer: 50 psi	Auxiliary gas: 60 psi
Nebulization temperature: 550 °C	Ingress voltage: +10 V

**Table 6 animals-14-02438-t006:** Standard curves and peak times of polyamines.

Items	Molecular Weight	Peak Time (Minutes)	Standard Curve	R
Putrescine	88.15	2.62	y = 755x + 589	0.9996
Spermidine	145.25	3.47	y = 567x + 788	0.9921
Spermine	202.34	4.41	y = 2.68 × 10^3^x + 5.72 × 10^3^	0.9912

**Table 7 animals-14-02438-t007:** Effects of *L*-citrulline supplementation on the metabolism concentrations of plasma amino acids in *Yili* horses (*n* = 6, nmol/mL).

Item	Control Groups	*L*-citrulline Groups	*p*-Value
Citrulline	75.63 ± 10.10 ^b^	94.73 ± 17.70 ^a^	<0.05
Arginine	79.60 ± 17.05 ^b^	101.00 ± 15.77 ^a^	<0.05
Ornithine	56.40 ± 15.93	54.43 ± 12.14	0.815
Essential amino acids			
Valine	159.80 ± 26.99	155.7 ± 21.61	0.774
Lysine	77.67 ± 27.97	98.03 ± 20.80	0.183
Threonine	98.97 ± 21.62 ^a^	72.63 ± 16.77 ^b^	<0.05
Leucine	87.33 ± 15.38	78.98 ± 16.34	0.384
Tryptophan	52.00 ± 7.13	51.88 ± 7.96	0.979
Phenylalanine	48.83 ± 5.94	44.43 ± 2.71	0.143
Isoleucine	41.47 ± 11.81	40.52 ± 10.12	0.884
Methionine	30.50 ± 7.93	28.70 ± 1.72	0.608
Essential amino acids	596.60 ± 111.60	570.90 ± 80.80	0.657
Non-essential amino acids			
Glycine	554.20 ± 47.35 ^A^	392.50 ± 42.42 ^B^	<0.001
Glutamine	273.00 ± 25.55	300.50 ± 19.97	0.065
Alanine	212.50 ± 24.97 ^a^	174.70 ± 29.31 ^b^	<0.05
Serine	167.20 ± 17.66 ^a^	136.33 ± 21.73 ^b^	<0.05
Proline	111.20 ± 20.01	95.38 ± 8.79	0.107
Histidine	84.55 ± 13.84	72.75 ± 13.91	0.172
Tyrosine	62.60 ± 10.09	59.88 ± 6.91	0.598
Cysteine	42.53 ± 13.20	45.67 ± 12.11	0.677
Glutamic acid	22.85 ± 3.51	23.42 ± 4.46	0.812
Asparagine	19.28 ± 4.94	20.10 ± 2.61	0.728
Taurine	12.10 ± 4.18	10.52 ± 3.43	0.491
3-Methyl-histidine	8.70 ± 3.99	9.03 ± 1.21	0.848
Pipecolic acid	7.51 ± 0.96	7.49 ± 1.08	0.972
Carnosine	7.41 ± 1.39	7.28 ± 0.69	0.835
Aspartic acid	6.55 ± 1.27	6.44 ± 1.66	0.907
4-Hydroxy-proline	5.91 ± 1.73	6.03 ± 1.92	0.913
Kynurenine	2.97 ± 0.24	3.31 ± 0.32	0.067
5-Hydroxy-tryptophan	2.34 ± 0.01	2.35 ± 0.011	0.114
β-Alanine	2.24 ± 0.41	1.95 ± 0.12	0.135
Ethanolamine	1.78 ± 0.49	2.05 ± 0.53	0.378
Homoarginine	1.51 ± 0.44	1.64 ± 0.21	0.502
5-Hydroxy-Tryptamine	1.27 ± 0.033	1.26 ± 0.015	0.450
2-Aminobutyric acid	1.40 ± 0.28 ^a^	1.09 ± 0.13 ^b^	<0.05
5-Hydroxylysine	1.23 ± 0.26	1.18 ± 0.56	0.854
α-Aminoadipic acid	0.97 ± 0.13	0.97 ± 0.25	0.999
Sarcosine	1.03 ± 0.37	0.92 ± 0.34	0.603
3-Aminoisobutyric acid	0.87 ± 0.11	0.83 ± 0.15	0.636
Homocitrulline	0.54 ± 0.14	0.53 ± 0.03	0.974
Cystathionine	0.41 ± 0.14	0.43 ± 0.14	0.776
Homoserine	0.38 ± 0.053	0.34 ± 0.023	0.135
5-Aminovaleric acid	0.20 ± 0.18	0.14 ± 0.042	0.408
Phospho-serine	0.12 ± 0.028	0.12 ± 0.032	0.709
Phosphorylethanolamine	0.061 ± 0.002 ^a^	0.0588 ± 0.00 ^b^	<0.05
Non-essential amino acids	1829.00 ± 171.40	1637.3 ± 148.1	0.065
TAA	2425.60 ± 270.80	2208.10 ± 221.3	0.159

Note: The values are expressed as means ± SD. Values with no letter or the same superscript letter within the same row do not significantly differ (*p* > 0.05). Values with different lowercase superscript letters within the same row significantly differ (*p* < 0.05). Values with different uppercase superscript letters within the same row very significantly differ (*p* < 0.01).

**Table 8 animals-14-02438-t008:** Effects of *L*-citrulline supplementation on plasma polyamine concentrations in *Yili* horses (*n* = 6, ng/mL).

Item	Control Groups	*L*-citrulline Groups	*p*-Value
Putrescine	36.10 ± 8.70 ^b^	50.93 ± 8.26 ^a^	<0.05
Spermidine	20.93 ± 5.81 ^b^	29.45 ± 6.98 ^a^	<0.05
Spermine	97.47 ± 26.21	109.30 ± 23.32	0.428

Note: The values are expressed as means ± SD. Values with no letter or the same superscript letter within the same row do not significantly differ *(p* > 0.05). Values with different lowercase superscript letters within the same row significantly differ (*p* < 0.05).

## Data Availability

The data that support the findings of this study are available from the corresponding author upon reasonable request.

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
