# Peer review of "The Effects of L-citrulline Supplementation on the Athletic Performance, Physiological and Biochemical Parameters, Antioxidant Capacity, and Blood Amino Acid and Polyamine Levels in Speed-Racing Yili Horses"

_animals, 2024, doi:10.3390/ani14162438_

Round 1
Reviewer 1 Report
Comments and Suggestions for Authors
high exercise- should be strenuous exercise or high intensity
exhibited a nonsignificant upward trend.
Doesn't an increase in creatine indicare kidney
84 speed not exercise level
87 in the plasma
Methods The horses were fed grain twice a day, but the citrline was added to only one meal Morning or evening feedin
Table 1 Does V stand for vitamin? Usually abbreviated Vit B1
139 stethoscope on the horse's chest
Fig 1 Derivatization is not familiar to this reviewer Derivation ?
Fig 2 Respiratory rate not breath rate Nothing significant. Is that correctord
Through out the methods the wording should be changes decreased or increased to was higher or lower than the control group espevcial;ly for the prerace dats 274 was higher not increased 300 increased from what 2 hours before the race or do you mean was lower than control group at that time.
Fig 4 blood routine ? what is that
Fig 5 define all initials in the figure legend T- AOC
366 misrepresented- don't you mean represented
376 putrescine, spermidine, and spermine What is the significance of the increase in amino acids that are associated with decay
386 give actual minute
454 between the control and experimental group
473 More damage in citrulline supplemented group
563 omit significantly
597 This does not make sense omit 3 hours or add 3 hours after
617 in the citrulline supplemented group
Comments on the Quality of English Language
See above
Author Response
Thank you for taking time out of your busy schedule to review the manuscript and for your valuable comments and suggestions. I have responded to your comments one by one. Please refer to the attached document.

Reviewer 2 Report
Comments and Suggestions for Authors
REVIEW
"Effects of L-Citrulline Supplementation on the Athletic Performance, Physiological and Biochemical Parameters, Antioxidant Capacity, and Blood Amino Acid and Polyamine Levels in Speed-Racing Yili Horses"
BRIEF SUMMARY AND GENERAL COMMENTS:
The present study evaluated a supplement with the potential to enhance horses' athletic performance and post-workout recovery. L-citrulline has indicated potential for sports use in humans, but it is still poorly studied in horses.
The study was comprehensive, investigating various aspects of l-citrulline supplementation.
However, in some aspects of the Discussion and conclusions, understandings were described that needed to be more consistent with the statistics presented, requiring rewriting a good part of the Discussion and conclusions and part of the simple summary and abstract.
SPECIFIC COMMENTS:
Simple summary
It is well-written, direct, and objective. Please refer to the comments in the Discussion and conclusions about athletic ability and post-exercise recovery.
Abstract
Well written, clear and objective.
Line 22: Articles should be written in the past tense, reporting what has been accomplished. Please change the word "is" to "was"
Please refer to the comments in the Discussion and conclusions about athletic ability and post-exercise recovery.
It is important to distinguish the groups at the 'end of the race' (referring to time zero) from the groups "after the race" (referring to 2 hours after the effort), both here and throughout the text.
Introduction
It is well structured and presented.
Materials and Methods
Line 107 and 112: Was the supplement offered at a time independent of the ration, or did the time of supply of the night supplement or the 19 ration occur jointly? If it occurred jointly, please adjust one of the schedules.
Line 113: The nutritional part is very well presented. It could be even better to detail the roughage offered. For example, what was the roughage offered, and possibly what was its nutritional composition?
Line 115: Consider creating a new paragraph from "After"
Table 1 does not clarify whether 'dry forage' refers to the total forage received throughout the day or whether it is the fibrous matter of the concentrate. It would be worth the work to detail this aspect; the L of soybean meal probably is typed with a capital letter by mistake.
Line 140: Consider creating a new paragraph from "For"
Line 145: Consider creating a new paragraph from "On"
Line 146: Consider joining sentences by putting "," after "conducted"
Line 152: Consider creating a new paragraph from "These"
Line 158: Please check for "," after "Co" and after "Ltd." Please insert space before the parentheses.
Lines 173 to 184: The text is presented as a list. Please rewrite it in more detail using scientific language to clarify the information presented.
Lines 186-195: The verb tense to describe the methodology should be the past tense, reporting what was accomplished. Please correct the verb tenses in this segment in this sense.
Lines 197-208: The text is presented as a list. Please rewrite it in more detail, using scientific language, to clarify the information presented.
Line 220: Please include some references supporting the methodology used.
Line 225: Please include some references supporting the methodology used.
Line 233: Please include some references supporting the methodology used.
Line 239: Please include some references supporting the methodology used.
Line 241: Is the word "collation" correct?
Results
Consider presenting the figures in the same sequence in which they are presented in the text. When using parentheses, be sure to insert a space before them. When addressing each topic, it is recommended to prioritize the parameters that showed a significant result first, followed by those that approached significance and those that were not significant.
Line 253-257: Please include the statistical values, if possible.
Line 257: Please refer to Figure 2A
Line 262: Please consider removing the word "however"
Line 273: Consider spelling terms like "Glu" in full, making the text more accessible to people unfamiliar with these terminologies.
Line 282: Please include the statistical values
Line 331: "MDA" is possibly actually "MIDA".
Line 355: The word "that" does not seem necessary in the sentence.
Lines 355-359: PCA analysis (The Figure would need to be explained in more detail. If it does not add relevant information, evaluate deleting it.
Lines 359-363: The Pearson correlation (Figure 6C) analysis would need to be explained in greater detail. If it does not add relevant information, evaluate its deletion. To explain it better, highlight where citrulline is found in the graph.
Line 359: It appears that 5C is 6C.
Table 4. It is recommended that consistent technical terminology be used throughout the document, such as "Trial groups" versus "Experimental groups".
Discussion
The topic would be more precise if it followed the pattern adopted in the results section.
Research on citrulline supplementation in horses is sparse, which justifies comparison with other species, such as humans and others. However, whenever results from other species are presented, it is necessary to contextualize and inform the species evaluated. For each topic of Discussion, first address results obtained with horses, followed by closer species, such as sheep, and then species further away from the equine in terms of physiology.
Lines 387-390: This statement is not statistically accurate. The absence of statistical significance indicates that the results obtained were random. Thus, if the experiment is repeated, the results may differ from those obtained here. For this reason, it is not possible to make conjectures.
Line 391: As detailed above, in the absence of statistical difference, it is not statistically correct to state that there has been an improvement. This Discussion could be placed within the scope of hypotheses, always presenting values of the observed significance.
Line 404-405: Please present some references.
Line 413: Refer to the Figure that shows this result. Follow this pattern throughout the text.
Lines 413-415: The difference between groups should be discussed rather than the difference between periods.
Line 415-418: The data do not allow this statement since, after 2 hours, there was no difference between the groups concerning lactate, indicating a good recovery in both groups. Whether the degree of exertion was sufficient to induce differences in lactate 2 hours post-exercise could be questioned.
Lines 537-538: Please present some references.
Lines 562-564: In Figure 5F, there was no statistical difference. Thus, describing this result as "tends to significantly increase plasma NO concentration" is inaccurate in statistical terms.
Line 574: "was" would be better than "is".
Lines 581-582: Please present some references.
Line 601-602: If there was no statistical difference, it is not correct to state that the concentration of ornithine was different between the groups, even though numerically, the difference was large. Please correct similar statements throughout the text.
Conclusions
The conclusions regarding the adjustments in the Discussion would need to be adjusted. For instance, you presented statistically similar values regarding racing time and lactate concentration 2 hours post effort.
Author Response

(The authors gave the same response as above.)

Reviewer 3 Report
Comments and Suggestions for Authors
To The Authors:
Thank you for submitting this very interesting study. I think overall the data gained in this study is very useful! Overall I think this study can be presented in a much more succinct manner.
Introduction:
66: I would note that these are human studies
Figure 1: The image is a bit distorted on the version that I have
Methods:
Was there a control for jockey? Or at least weight of jockey? If not this needs to be listed as a limitation as rider can have a major impact on overall time performance.
Statistics: What was the rational for not using an ANOVA? I guess I would have compared pre-race, post-race and 2 hours post for each group. It would have given the data more impact.
Results:
3.1/3.111 : I do not think both titles are needed
Figure 2: There is a misspelling of Figure. I would also have the axis as respiratory rate vs breath rate.
Figure 3, 4, 5,6: Figure is misspelled
Table 4: What do you mean by data is misrepresented? If it is misrepresented, then I would think it should be taken out. The letter system is very confusing. I think you could just put a star after the p value. Or in the legend say p was set at X for significance.
Discussion
Overall, the discussion is VERY long and a bit hard to follow. I would suggest condensing the information to focus on your findings. You have a large amount of information about pathology but I am not sure that it is overall relevant to your findings. I would overall restructure this discussion starting with your major findings. I have made some suggestions on things that could potentially be removed though they are just suggestions!
430: Unless your blood gas was arterial you cannot really make a statement about this. Though VBG is useful you really need arterial to determine degree of hypoxia. I think this should be listed as a limitation somewhere.
454: This sentence is very confusing. What do you mean by gas metabolism?
461-471: I Think you can remove all of this, maybe see what the other reviewers say but I am not sure this information contributes.
474: I don’t think you can say this. You can say the number of monocytes was significantly decreased two hours before. That being said did you normalize your data? It is possible that your test group just had a lower circulating pool of monocytes physiologically. I think you would need to compare to pre-treatment values if you wanted to make this statement.
496: I think this information is confounded by the fact your HCT went up immediately after and hemoconcentration can have an effect on TP. Additionally I question this comparison. I think it would have been more useful for these markers to use an ANOVA

There are a few grammatical errors which can be fixed in production.
Author Response

(The authors gave the same response as above.)

Round 2
Reviewer 2 Report
Comments and Suggestions for Authors
Thank you very much for the acceptance of the introduced suggestions. Substantial adjustments and improvements were made to the manuscript, satisfactorily addressing the points placed on the previous evaluation. The writing became clearer, data was enhanced and presented in a better manner, new references enriched the discussion. The article has gone through proper organization, has a clear line of development, and is ready to be published.
Some minor corrections should be made:
Line 19: Changing “S” by “s”.
Line 39: Changing “the” by “The”
Lines 242 and 247: Changing “figure” by “Figure”
Reviewer 3 Report
Comments and Suggestions for Authors
To the authors:
Thank you for addressing my comments. The manuscript is improved from the initial submission and is acceptable for publication.